# Antioxidant Peptides from the Collagen of Antler Ossified Tissue and Their Protective Effects against H_2_O_2_-Induced Oxidative Damage toward HaCaT Cells

**DOI:** 10.3390/molecules28196887

**Published:** 2023-09-30

**Authors:** Xi Chen, Peijun Xia, Shuo Zheng, Yi Li, Jiayuan Fang, Ze Ma, Libo Zhang, Xunming Zhang, Linlin Hao, Hong Zhang

**Affiliations:** 1College of Animal Science, Jilin University, 5333 Xi’an Road, Changchun 130062, China; xic21@mails.jlu.edu.cn (X.C.); pjxia20@mails.jlu.edu.cn (P.X.); zhengshuo22@mails.jlu.edu.cn (S.Z.); yil22@mails.jlu.edu.cn (Y.L.); jyfang21@mails.jlu.edu.cn (J.F.); maze21@mails.jlu.edu.cn (Z.M.); zlb22@mails.jlu.edu.cn (L.Z.); xunming22@mails.jlu.edu.cn (X.Z.); 2Hospital of Stomatology, Jilin University, Changchun 130015, China

**Keywords:** antler ossified tissue, collagen peptide, antioxidant activity, oxidative damage

## Abstract

Antler ossified tissue has been widely used for the extraction of bioactive peptides. In this study, collagen was prepared from antler ossified tissue via acetic acid and pepsin. Five different proteases were used to hydrolyze the collagen and the hydrolysate treated by neutrase (collagen peptide named ACP) showed the highest DPPH radical clearance rate. The extraction process of ACP was optimized by response surface methodology, and the optimal conditions were as follows: a temperature of 52 °C, a pH of 6.1, and an enzyme concentration of 3200 U/g, which resulted in the maximum DPPH clearance rate of 74.41 ± 0.48%. The peptides (ACP-3) with the strongest antioxidant activity were obtained after isolation and purification, and its DPPH free radical clearance rate was 90.58 ± 1.27%; at the same time, it exhibited good scavenging activity for ABTS, hydroxyl radical, and superoxide anion radical. The study investigated the protective effect of ACP-3 on oxidative damage in HaCaT cells. The findings revealed that all groups that received ACP-3 pretreatment exhibited increased activities of SOD, GSH-Px, and CAT compared to the model group. Furthermore, ACP-3 pretreatment reduced the levels of ROS and MDA in HaCaT cells subjected to H_2_O_2_-induced oxidative damage. These results suggest that collagen peptides derived from deer antler ossified tissue can effectively mitigate the oxidative damage caused by H_2_O_2_ in HaCaT cells, thereby providing a foundation for the utilization of collagen peptides in pharmaceuticals and cosmetics.

## 1. Introduction

Oxidation of biomolecules has been identified as a physiological process mediated by free radicals. During normal metabolism in the human body, large amounts of reactive oxygen species (ROS) including superoxide radicals (O^2−^), hydrogen peroxide (H_2_O_2_), hydroxyl radicals (OH^−^), and nitroxyl radicals (NO), would be produced [1,2,3]. When the excessive production of reactive oxygen species or disruption of the free radical scavenging system in the body occurs, the dynamic balance between the production and scavenging of free radicals is severely disturbed, thus resulting in the accumulation of free radicals. Excessive free radicals can disrupt the normal structure of biomolecules in the body, which in turn leads to aging and diseases such as Alzheimer’s disease, cancer, and cardiovascular diseases [3,4]. Therefore, antioxidant supplementation plays a vital role in human health. Most of the synthetic antioxidants currently being used in large quantities, such as butylated hydroxytoluene (BHT) and butylated hydroxyanisole (BHA), have potential toxicity risks [3]. In recent years, many studies have reported that certain peptides have the ability to scavenge free radicals and prevent peroxidation in the body, and peptide antioxidants have the advantage of being relatively safe and well-tolerated. However, there are a few types of naturally occurring antioxidant peptides. Therefore, the research on making hydrolysis of parental proteins, such as collagen, to obtain antioxidant peptides is gradually becoming a hot topic.

Collagen constitutes the principal protein in animal bones and connective tissues, accounting for approximately 30% of the total protein in animals and serving as the most abundant protein in mammals [5,6]. At least 29 types of collagen have been identified, which have different amino acid compositions and sequences to perform different biological functions in the organism [7,8]. Recent studies have shown that collagen hydrolyzates may have high antioxidant activity [9]. Collagen can be used as a source of antioxidant peptides. Traditional sources of collagen are porcine skin, bovine skin, and bone. The use of collagen and products from porcine and bovine sources has been limited due to the influence of Judaism and Islam religious beliefs [10,11]. The substitutes such as marine fish have the potential risk of heavy metal accumulation [12,13]. As ruminants, the use of deer is less restricted by religion. Pepłowska K et al. [14] determined that the best condition for obtaining collagen yield was to degrease antlers in butanol at 25 °C, and then extract them in acetic acid for 36 h. The optimized collagen yield was 22.04%; Lee et al. [15] prepared acetic acid-soluble collagen (ASC) and pepsin-soluble collagen (PSC) from the antler velvet of elk. The extraction method of ASC was as follows: remove the collagen with 1:10 (*w*/*v*) solid-liquid ratio for 3 days, then degrease it with 10% butyl alcohol and 1:10 (*w*/*v*) solid solvent for 24 h, and extract it with 0.5 M acetic acid for 3 days. The ASC yield of 5.2% was obtained. The extraction method of PSC was as follows: 10% pepsin was added to the remaining solution and hydrolyzed for 48 h. The supernatant was collected by centrifugation of the viscous solution at 4 °C and 12,000× *g* for 1 h. The supernatant was salted out by adding NaCl with a final concentration of 0.7 M into 0.05 M Tris-HCl (pH 7.5) solution and then precipitated by adding NaCl with a final concentration of 2.3 M. The obtained precipitation was dissolved in 0.5 M acetic acid and dialyzed in 0.1 M acetic acid distilled water. The ASC yield of 6.91% was obtained. Therefore, deer antler is well suited as a raw material for collagen, providing a source of antioxidant peptides.

As captive-bred deer are already raised in large numbers in China, Russia, New Zealand, and different parts of Western Europe for their meat and hides, the antler is a readily available bioresource. As the only fully renewable complex organ in mammals, antlers are shed and regrown every year after the deer rut [16]. Antlers’ growth rates can be as fast as 1–3 cm per day [17]. The acquisition of antlers does not imply the killing of the carrier, making antlers a renewable biological resource. The annual production of antlers in China is about three hundred tons, more than one-third of which is the ossified part with a low medicinal value. The ossified part of the antler is difficult to use as medicinal material because it contains fewer growth factors and other substances, which causes a large waste. The collagen content of the antler increases with the degree of ossification, thus the collagen content of the ossified part of the antler is the highest, which is suitable for the extraction of antioxidant collagen peptides [18].

Previously, collagen extraction from antlers has been studied. However, there was no information available about the extraction of collagen peptides from the antler ossified part. Therefore, our study further utilized the antler resources and obtained products with the highest DPPH radical clearance using neutral protease combined with response surface methodology optimization. After ultrafiltration and molecular sieve chromatography, peptides (ACP-3) with the highest DPPH free radical clearance were obtained, and the in vitro antioxidant activities of the ACP-3 were evaluated, especially their cytoprotective effects on H_2_O_2_-damaged HaCaT cells.

## 2. Results and Discussion

### 2.1. Protease Selection and Single-Factor Experiments

The antioxidant properties of enzymatic hydrolysates of AC were evaluated by measuring their DPPH clearance rates. Five proteases (Neutrase, Alcalase, Papain, Trypsin, and Flavourzyme) were applied to hydrolyze AC and the resulting hydrolysates were compared (Figure 1A). Neutrase hydrolysate showed a significantly higher DPPH clearance rate than the other four protease hydrolysates. This result agrees with previous reports on the antioxidant peptides of Monkfish (*Lophius litulon*) [19] and Swim Bladders of Giant Croaker (*Nibea japonica*) [1] obtained by Neutrase hydrolysis. Protease specificity has been shown to influence the characteristics of the antioxidant peptides produced by hydrolysis [1]. Therefore, Neutrase was selected as the optimal protease for further experiments.

The effects of five single factors (hydrolysis time, enzyme dosage, substrate concentration, pH, and hydrolysis temperature) on the DPPH clearance rate of Neutrase hydrolysate were investigated (Figure 1B–F). The DPPH clearance rate increased and then decreased with the variation of each factor level within the experimental range. The optimal levels of the five factors for obtaining the highest DPPH clearance rate were a hydrolysis time of 3 h, an enzyme concentration of 3000 U/g, a concentration of substrate of 10%, a pH value of 6.0, and a hydrolysis temperature of 50 °C, respectively.

### 2.2. Optimization of Hydrolysis Conditions by Response Surface Methodology (RSM)

The Box-Behnken Design (BBD) and RSM were utilized to establish the ideal enzyme dosage, reaction temperature, and pH. Table 1 presents the experimental design and results. A second-order polynomial equation was employed to forecast the DPPH radical scavenging rate, considering the impact of these three variables. The function is given by:$$R\;(\%)=73.16+0.85A+2.45B+1.44C+0.052AB-1.05AC-0.49BC-1.75A^{2}-2.68B^{2}-2.33C^{2}$$
where R is the DPPH radical scavenging rate, *A* is the enzyme dosage, *B* is the temperature, and *C* is the pH.

The analysis of variance (ANOVA) for the regression model using F-test and *p*-test is shown in Table 2. The statistical model proved significant as indicated by the *p*-value of <0.0001. Both linear and quadratic terms of variables A, B, C, and D exhibited high significance in the model, and the interaction terms AC and BC were significant. The R^2^ of 0.9949 indicated a high correlation between the predicted and observed values of the model, and the R_adj_^2^ of 0.9884 implied that the model could explain 98.8% of the variation in the DPPH radical scavenging rate of Neutrase hydrolysate. The lack of fit term was not significant as its *p*-value was greater than 0.05, suggesting that the experimental error was minimal, the model fit was excellent and the model predictions were highly consistent with the observed values.

The Box-Behnken test regression model was utilized to obtain each response surface in this study. The three-dimensional model of the response surface and the contour plot’s center point provided the maximum response values under the interaction. As illustrated in Figure 2, the DPPH clearance rate gradually increased with the rise of temperature from 45 °C to 50 °C, and reached its peak value near 50 °C. Within the optimal temperature range, the enzyme efficiently digested the substrate, resulting in high biological activity for both the enzyme and enzymatic products. The DPPH clearance rate increased initially and then decreased with changes in pH, indicating that high or low pH adversely affected enzyme activity. An excessive dosage of enzymes may also result in a reduced DPPH clearance rate, possibly due to excessive hydrolysis caused by an excessive amount of enzymes. The response surface plots indicated that the highest DPPH radical scavenging rate under all three interactions was appropriately selected.

Based on the results of the regression model, the optimal enzymatic digestion process of AC was determined. The theoretical optimal enzymatic digestion process entailed an enzyme dosage of 3180.79 U/g, reaction temperature of 52.19 °C, pH of 6.11, and a theoretical DPPH scavenging rate of 73.94% for the enzymatic product. The enzymatic process was then adjusted, with an enzyme dosage of 3200 U/g, a reaction temperature of 52 °C, a pH of 6.1, a substrate concentration of 10%, and an enzymatic time of 3 h. To validate the process, three repetitions were conducted, and the final DPPH clearance rate was 73.41 ± 0.48%, which closely matched the theoretical value.

### 2.3. Molecular Weight Distribution of ACP

Previous studies have demonstrated a close relationship between the antioxidant activity of peptides and their molecular weight, with smaller peptides (< 3 kDa) often exhibiting stronger antioxidant activity [20]. Here, we employed high-performance liquid chromatography to determine the relative molecular weight distribution of ACP. Our linear regression analysis of relative molecular mass revealed an equation of lg(M) = −0.2414Rt + 6.997, R^2^ = 0.9889. As shown in Figure 3, multiple fractions with different relative molecular masses were detected in ACP, with 99.6% of these products possessing molecular masses less than 10 kDa. Furthermore, over 98% of the enzymatic digestion products had molecular weights less than 3 kDa and the proportion of products less than 1 kDa was found to be 82.85%. These findings suggest that small-molecule polypeptides comprise a major proportion of ACP, indicating their high antioxidant potential.

### 2.4. Isolation and Purification of ACP

To fractionate the enzymatic digestion products based on their molecular weights, an ultrafiltration centrifugation technique was employed to separate ACP into three fractions: UF-1 (>10 kDa), UF-2 (3–10 kDa), and UF-3 (<3 kDa). The DPPH clearance rate of each fraction and the original enzymatic digestion products before ultrafiltration were assessed separately, as depicted in Figure 4A. The yield of UF-1, UF-2, and UF-3 accounted for 2%, 4%, and 94% of ACP, respectively. The DPPH clearance rate for the UF-3 fraction was 83.36 ± 0.62%, and there was a significant difference in the DPPH clearance rate among the three fractions, showing a trend of higher scavenging rate with decreasing molecular weight. According to the literature, we know that the parts with the strongest antioxidant activity are composed of components with low molecular weight. Consequently, the enzymatic digestion product in the UF-3 fraction was selected for further isolation and purification experiments. Figure 4B shows the results of the isolation and purification of UF-3 using a Sephadex G-25 gel column. Four fractions were obtained, namely ACP-1, ACP-2, ACP-3, and ACP-4. The yield of ACP-1, ACP-2, ACP-3, and ACP-4 accounted for 12%, 11%, 45%, and 32% of UF-3, respectively. The DPPH clearance rate of each fraction was determined separately, and it was found that ACP-3 had the highest DPPH clearance rate of 90.58 ± 1.27%, which was significantly higher than the other three fractions. Therefore, ACP-3 was collected.

### 2.5. Amino Acid Composition of ACP-3

Studies have shown that antioxidant peptides can directly eliminate ROS through hydrogen or electron donation. The hydrogen or electron-donating role of antioxidant peptides depends on their amino acid residues. The aromatic amino acids tyrosine (Tyr), tryptophan (Trp), and phenylalanine (Phe) can provide hydrogen protons to electron-deficient free radicals and maintain stability through resonance structures. The chelation of peptides with metal ions plays an antioxidant role by altering the chemical reactivity of metals, forming insoluble metal complexes, or spatially hindering metal-lipid interactions to prevent the formation of free radicals. In this study, the amino acid composition of ACP-3 was determined to further understand its antioxidant properties. As shown in Table 3, a total of 17 amino acids were detected by ACP-3, including 7 essential amino acids and 10 non-essential amino acids. The proportion of hydrophobic amino acids in ACP-3 was 30.9%, the proportion of acidic amino acids was 17.43%, and the proportion of basic amino acids was 14.57%. The most abundant amino acids in ACP-3 are Pro and Gly, which is similar to the previous studies on collagen peptides, indicating that ACP-3 belongs to collagen peptides [21]. Pro has been proven to be closely related to antioxidant capacity in many studies, and it is mainly believed that the pyrrole ring in Pro residues can chelate metal ions by providing protons to achieve the effect of scavenging free radicals. In addition to Pro, hydrophobic amino acids in ACP-3 include Ala, Met, Ile, Leu, and Phe. Hydrophobic amino acids are believed to maintain the stability of the peptide at the water-lipid interface and help to remove free radicals generated by the lipid phase [22]. Therefore, peptides containing more hydrophobic amino acid residues tend to have better antioxidant properties. ACP-3 contains basic amino acids such as His, Lys, Arg, etc., which can remove free radicals and prevent oxidation by providing hydrogen atoms or chelating with metal ions with opposite charges [23]. Acidic amino acids such as Glu and Asp, which are rich in ACP-3, can be used as reducing agents to remove free radicals because of their negative charge and can inhibit the pro-oxidation reaction by chelating with metal ions. The contents of hydrophobic, acidic, and basic amino acids in ACP-3 are higher, which indicates that ACP-3 has strong antioxidant activity.

### 2.6. DPPH, ABTS, Hydroxyl Radical, and Superoxide Anion Clearance Rate of ACP-3

In order to evaluate the antioxidant activity of ACP-3, various assays were conducted to measure the clearance rates of different free radicals, including DPPH, ABTS, hydroxyl radicals, and superoxide anion radicals. ACP-3 showed dose-dependent scavenging capabilities of DPPH, ABTS, hydroxyl radicals, and superoxide anion radicals (Figure 5A–D), which indicates that ACP-3 is a potential component to be used as an antioxidant for pharmaceuticals and cosmetics.

### 2.7. Cytotoxicity of ACP-3 at Different Concentrations on HaCaT Cells

The cytotoxicity of HaCaT cells was assessed by the CCK8 method. As shown in Figure 6, HaCaT cells were exposed to various concentrations of ACP-3 for 24 h, and the cell viability did not decrease. Thus, ACP-3 does not exhibit toxicity at normal administration concentrations and has a strong potential for oxidative stress.

### 2.8. Protective Effect against H_2_O_2_-Induced Damage on HaCaT Cells

In this study, the cytotoxicity of ACP-3 at different concentrations to HaCaT cells was investigated by the CCK8 method to determine the appropriate dosage of ACP-3. The results, as depicted in Figure 7, demonstrated that the cell viability of H_2_O_2_ model groups decreased to 47.22% of the control group, and the cell viability of H_2_O_2_ model groups increased to 62.42% of the control group after 50 μg/mL ACP-3 pretreatment, which was significantly different from the model group. Further increasing the concentration of ACP-3 to 200 μg/mL restored the cell viability to 90.12%, which was significantly different from the model group, but not significantly different from the control group. When ACP-3 concentration continued to increase, cell viability was no longer significantly improved. In conclusion, when ACP concentration is greater than 50 μg/mL, ACP shows a significant protective effect on H_2_O_2_-induced damage of HaCaT cells.

### 2.9. Effect of ACP-3 on ROS Content with H_2_O_2_-Induced Damage on HaCaT Cells

ROS is in a state of constant balance between production and elimination in the body, and this balance is necessary for the regulation of physiological activities in the body [24]. The level of intracellular ROS can directly reflect the level of oxidative damage of cells [25]. As shown in Figure 8, When HaCaT cells were treated with 500 μM H_2_O_2_, the intracellular ROS level was significantly higher compared to the control group. After treatment with different concentrations of ACP-3, the fluorescence intensity decreased significantly, and the reduction amplitude increased with the increase of ACP-3 concentration, indicating that ACP-3 has a dose effect on the regulation of ROS levels. These results indicate that ACP-3 could effectively prevent ROS accumulation and help cells resist oxidative damage.

### 2.10. Effects of ACP-3 on the Levels of SOD, CAT, GSH-Px, and MDA with H_2_O_2_-Induced Damage on HaCaT Cells

SOD, CAT, and GSH-Px are a series of antioxidant enzymes that play a crucial role in removing free radicals generated in normal cells. MDA is the final product of cellular lipid peroxidation and can reflect the severity of cellular oxidative damage. To measure the antioxidant activity of ACP-3, we studied the effects of pretreatment with ACP-3 on SOD, CAT, GSH-Px, and MDA levels after H_2_O_2_ injury. As shown in Figure 9A–C, the levels of SOD, CAT, and GSH-Px in ACP-3 pretreated cells increased in a dose-dependent manner, and were significantly higher than those in H_2_O_2_-induced cells. Figure 9D showed that MDA content decreased with the increase of peptide concentration after the pretreatment of HaCaT cells with ACP-3. In conclusion, ACP-3 can reduce oxidative stress damage and inhibit lipid peroxidation.

## 3. Materials and Methods

### 3.1. Materials

Fresh antler ossified tissue was supplied by Jilin Changsheng Deer Industry Co., Ltd. (Jilin, China) and stored at a temperature of −20 °C prior to use. Trypsin (250 U/mg), neutrase (100 U/mg), alcalase (200 U/mg), pepsin (3000 U/mg), and papain (800 U/mg) were purchased from Shanghai yuanye Bio-Technology Co., Ltd. (Shanghai, China). HaCaT cells were gifted from Zhou Liting, School of Public Health, Jilin University [26]. 

### 3.2. Preparation of Collagen

The extraction of collagen was carried out as described by Wu et al. with minor modifications [27]. Frozen antler bones were crushed in a pulverizer after being cleaned and skinned. Powdered antler bones were mixed with a solution of 0.1 mol/L NaOH in a ratio of 1:10 (*w*:*v*). The mixture was stirred continuously with a magnetic stirrer at 200 r/min for 12 h. The solution of alkali was changed every 6 h to remove the non-collagenous protein. After washing with water to neutral, the antlers were further demineralized in a solution of 0.5 mol/L EDTA-2Na (pH 7.4) in a ratio of 1:10 (*w*:*v*). The mixture was continuously stirred for 48 h, with the solution being changed every 12 h. The antler bones were then rinsed with cold water until the washing solution was neutral. All preparations were carried out at 4 °C.

The cleaned antler bones were stirred for 72 h using 0.5 mol/L acetic acid containing 0.06% pepsin at a ratio of 1:20 (*w*/*v*) to extract the collagen. The mixture was filtered through two layers of gauze. NaCl was added to a final concentration of 2.4 mol/L at 0.05 mol/L tris (hydroxymethyl) aminomethane (pH 7.0) and left for 12 h at 4 °C to precipitate the collagen in the filtrate. The precipitate was obtained through centrifugation at 10,000× *g* for 60 min at 4 °C utilizing a refrigerated centrifuge. The precipitate was then dissolved in a minimal amount of 0.5 mol/L acetic acid and placed into dialysis bags (8000–14,000 Da, Solarbio). The dialysis process was carried out using 25 times the volume of 0.1 mol/L acetic acid for 24 h. The precipitate was subsequently subjected to dialysis with distilled water at a volumetric ratio of 25:1 for 48 h. The dialysate obtained after this process was freeze-dried to obtain antler collagen, identified as AC. All experiments were performed at 4 °C.

### 3.3. Optimization of Preparative Conditions of Collagen Hydrolysate from AC

AC was hydrolyzed by each of the five different proteases according to the designed conditions (Table 4). Afterward, the hydrolysis products (named ACP) were subjected to a 100 °C water bath for 10 min before being centrifuged at 6000× *g* for 10 min. The supernatant was collected to measure the DPPH Clearance Rate.

According to the results of the DPPH Clearance Rate, neutral protease was chosen as the optimum protease. Five parameters of enzyme concentration, temperature, enzymatic digestion time, pH, and substrate concentration were selected for the single-factor experiment with DPPH Clearance Rate as the index.

On the basis of the single-factor experiment, three parameters, enzyme concentration, temperature, and pH, were selected to design a three-level three-factor BBD using Design Export software. The test factors and levels are shown in Table 5. The following polynomial quadratic model was used to assess the relationship between the dependent and independent variables:$$\gamma =\beta_{0}+\overset{3}{\underset{i=1}{\sum }}\beta_{i}X_{i}+\overset{3}{\underset{i=1}{\sum }}\beta_{ii}X_{i}^{2}+\sum \overset{3}{\underset{i<j=1}{\sum }}\beta_{ij}X_{i}X_{j}$$

In the equation, the dependent variable is represented by *γ*, while the independent variables are represented by *X_i_* and *X_j_*. The regression coefficients for the intercept, linear, quadratic, and interaction terms are represented by *β_0_*, *β_i_*, *β_ii_*, and *β_ij_*, respectively.

### 3.4. Determination of the Mw Distribution of ACP

The molecular weight distribution of ACP was examined by employing a high-performance liquid chromatography system (Waters 2695, Burlington, MA, USA). The mobile phase comprised of 40% (*v*/*v*) acetonitrile, 60% (*v*/*v*) water, and 0.1% (*v*/*v*) trifluoroacetic acid, the flow rate was 0.5 mL/min, and the gel chromatography was calibrated using standard proteins such as cytochrome C (Mw12384), peptidase (Mw6500), bacillus peptide (Mw1422), ethanine-ethanine monotyrosine monoarginine (Mw451), and ethanine-ethanine-ethanine (Mw189) (Sigma-Aldrich, Shanghai, China) as a standard. The molecular weight distribution of ACP was calculated from the equation of the standard curve of molecular weight versus retention time.

### 3.5. Fractionation of ACP

The solution of ACP was graded according to molecular weight using ultrafiltration tubes (Millipore, Burlington, MA, USA) with 10,000 Da and 3000 Da MWCO. This process yielded three fractions: > 10,000 Da, 3000–10,000 Da, and < 3000 Da. They were named UF-1, UF-2, and UF-3, respectively. The graded fractions were lyophilized and UF-3 with the best DPPH Clearance Rate was selected for subsequent separation. Further separation of UF-3 was performed using gel filtration chromatography. A 5 mL sample of UF-3 solution (20.0 mg/mL) was loaded onto a Sephadex G-25 column measuring 2.6 cm × 80 cm, and eluted with distilled water at a flow rate of 0.75 mL/min. Each step was monitored at 280 nm, collected in a 5 mL volume, and lyophilized. The fractions ACP-1 to ACP-4 were then sorted based on their absorbance at 280 nm.

### 3.6. Amino Acid Composition Measurement of ACP-3

ACP-3 was dissolved in 6 mol/L HCl at 110 °C and hydrolyzed for 24 h. The sample was evaporated via blowing in nitrogen. The amino acid composition content was measured by the amino acid analyzer.

### 3.7. Antioxidant Activity of ACP-3

#### 3.7.1. DPPH Clearance Rate

The DPPH clearance rate assay is referenced to Wang et al. with some modifications [28], the specific method is as follows:

Samples of different concentrations were dissolved in distilled water. Then, 1.5 mL of the sample solution was combined with 1 mL of DPPH (0.1 mM) that was dissolved in ethanol. The mixture was left to stand for 30 min in the dark, and centrifuged at a rate of 4000× *g* for 5 min. The absorbance was measured at 517 nm. Deionized water served as the control group instead of the sample, and ethanol was used as the blank group instead of DPPH. The DPPH clearance rate was determined by employing the subsequent equation:$$DPPH\;clearance\;rate\;\left(\%\right)=\left(A_{c}+A_{b}-A_{s}\right)/A_{c}\times 100\%$$
where *A_s_* is the sample absorbance, *A_c_* is the control absorbance, and *A_b_* is the blank absorbance.

#### 3.7.2. ABTS Clearance Rate

The ABTS clearance rate assay is referenced to Stella et al. with some modifications [29], the specific method is as follows:

ABTS (7 mM) was mixed with potassium persulfate (2.45 mM) and left for 12 h to measure absorbance at 734 nm. ABTS (1 mL) was mixed with a sample (1 mL) and placed in darkness for 10 min to measure absorbance at 734 nm. In the control group, deionized water was used instead of the sample. The ABTS clearance rate was derived by applying the equation:$$ABTS\;clearance\;rate\;\left(\%\right)=\left(A_{c}+A_{b}-A_{s}\right)/A_{c}\times 100\%$$
where *A_s_* is the sample absorbance, *A_c_* is the control absorbance, and *A_b_* is the blank absorbance.

#### 3.7.3. Hydroxyl Clearance Rate

The hydroxyl clearance rate assay was according to to Cao et al. with some modifications [30], the specific method is as follows:

First, 100 μL salicylic acid (7.5 mM) and 100 μL FeSO4 (7.5 mM) were prepared and thoroughly mixed. The sample (100 μL) and 0.03% H_2_O_2_ were prepared, which were thoroughly mixed. The absorbance was measured at 536 nm after the incubation at 37 °C for 30 min. H_2_O was used to replace the sample in the control group, and H_2_O was used to replace the salicylic acid solution in the blank group. The hydroxyl clearance rate was calculated using the following equation:$$Hydroxyl\;clearance\;rate\;\left(\%\right)=\left(A_{c}+A_{b}-A_{s}\right)/A_{c}\times 100\%$$
where *A_s_* is the sample absorbance, *A_c_* is the control absorbance, and *A_b_* is the blank absorbance.

#### 3.7.4. Superoxide Anion Clearance Rate

The superoxide anion clearance rate assay is referenced to Chen et al. with some modifications [31], the specific method is as follows:

The reaction commenced by blending 1 mL each of nitrotetrazolium blue chloride (NBT) solution (2.52 mM) and NADH (624 mM), and 1 mL of the sample. Then, 1 mL of phenazine methosulfate (PMS) solution (120 μg) was added. After incubation for 5 min at room temperature, absorbance was measured at 560 nm. For the control group, deionized water was substituted for the sample. The rate of superoxide anion clearance was assessed via the below equation:$$Superoxide\;anions\;clearance\;rate\;\left(\%\right)=\left(A_{c}+A_{b}-A_{s}\right)/A_{c}\times 100\%$$
where *A_s_* is the sample absorbance, *A_c_* is the control absorbance.

### 3.8. Effects of ACP-3 on the H_2_O_2_-Induced HaCaT Cells

ACP-3 was dissolved in DMEM medium with the concentration of 25, 50, 100, 200, 400, 600, 800, and 1000 µg/mL, respectively. HaCaT cells (1 × 10^4^ cells/well) were cultured in 96-well plates for 24 h. Thereafter, 100 µL peptide samples were added to the protected groups and incubated for 24 h. After removing the peptide samples, 500 μM H_2_O_2_ was added to the damaged group and the protected group, and the mixed solution was incubated for 6 h. The 96-well plates were washed with phosphate buffer (PBS). 100 μL DMEM medium containing 10% CCK8 (*v*/*v*) reagent was added and incubated at 37 °C for 1 h. Absorbance was determined at 450 nm, and cell viability was calculated using the specified equation:Cell viability (%) = (*A_s_* − *A_b_*) / (*A_c_* − *A_b_*) × 100%
where *A_s_* is the sample absorbance, *A_c_* is the control absorbance, and *A_b_* is the blank absorbance.

### 3.9. Determination of the Levels of ROS in H_2_O_2_-Induced HaCaT Cells

HaCaT cells were seeded in 6-well plates at a cell density of 2 × 10^5^ cells/well. After incubation with the samples for 24 h, cells were incubated with 500 μM H_2_O_2_ for 8 h. Then, the culture medium was removed, rinsed three times with PBS, and incubated with fresh medium containing 10 μM fluorescent probe DCFH-DA for 0.5 h. Subsequently, cells were rinsed three times with PBS. The cell morphology was observed by fluorescence microscopy and photographed. The fluorescence intensity was analyzed by Image J software (J2x 2.1.4.7).

### 3.10. Determination of Antioxidant Enzyme and MDA Levels in H_2_O_2_-induced HaCaT Cells

HaCaT cells were seeded in 6-well plates at a cell density of 2 × 10^5^ cells/well. Different concentrations of samples were added to the protection group. 500 μL of cell lysate was added to each group and centrifuged at 12,000× *g* for 10 min at 4 °C. The resulting supernatant was cooled at 4 °C and set aside. The levels of superoxide dismutase (SOD), catalase (CAT), glutathione peroxidase (GSH-PX), and malondialdehyde (MDA) were quantified through commercially available kits obtained from Nanjing Jiancheng Institute of Biological Engineering in Nanjing, China. The results were reported in units of enzyme activity per mg protein (U/mg protein).

### 3.11. Statistical Analysis

The dataset is presented as the mean ± SD (*n* = 3). A one-way analysis of variance (ANOVA) test was conducted to assess the differences between group means. *p* < 0.05 was considered statistically significant.

## 4. Conclusions

In this investigation, ACP was extracted from collagen. Response surface methodology was utilized to optimize the ACP extraction process, resulting in the determination of the optimal conditions as follows: a temperature of 52 °C, a pH of 6.1, and an enzyme concentration of 3200 U/g. These conditions demonstrated a maximum DPPH clearance rate of 74.41 ± 0.48%. The peptide (ACP-3) with the strongest antioxidant activity was obtained after isolation and purification, and its DPPH free radical clearance rate was 90.58 ± 1.27%. At the same time, it exhibited favorable scavenging activity for ABTS, hydroxyl radical, and superoxide anion radical. Additionally, ACP-3 demonstrated the ability to protect HaCaT cells from oxidative stress induced by H_2_O_2_. These results lay a foundation for the further application of ACP-3 in pharmaceuticals and cosmetics.

## Figures and Tables

**Figure 1 molecules-28-06887-f001:**
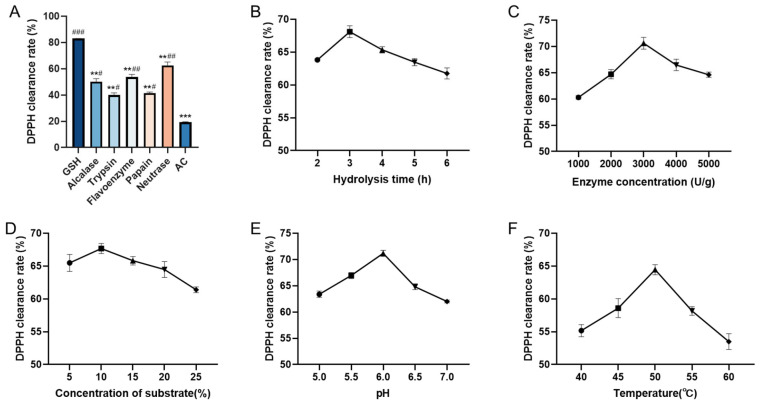
Effect of different conditions on the scavenging rate of DPPH radicals from enzymatic digestion products. (**A**). Different proteases; (**B**). hydrolysis time; (**C**). enzyme concentration; (**D**). concentration of substrate; (**E**). pH; (**F**). temperature. The concentration of enzymatic digestion products, GSH, and AC were 5 mg/mL. All results were triplicates of the mean ± SD. ** *p* < 0.01, *** *p* < 0.001 versus GSH; # *p* < 0.05, ## *p* < 0.01, ### *p* < 0.001 versus AC.

**Figure 2 molecules-28-06887-f002:**
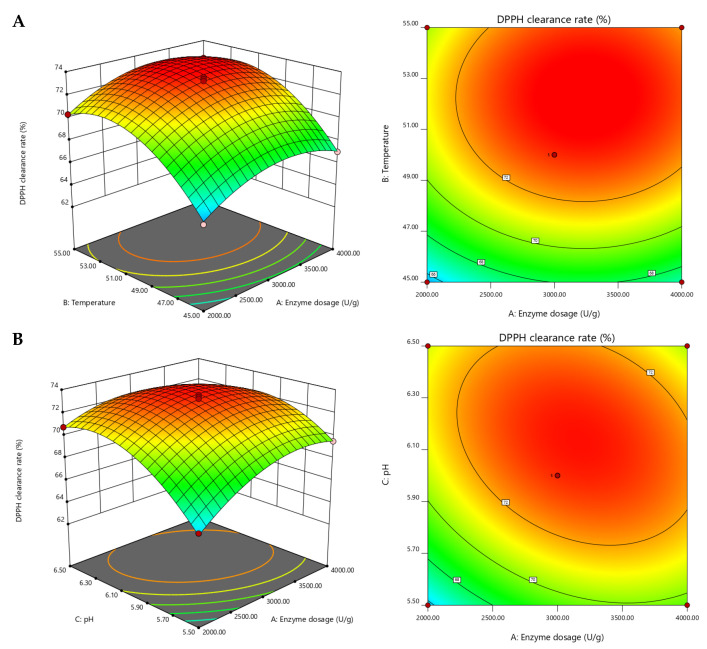
Response surface of DPPH clearance rate. Images represent the following: (**A**). effects of enzyme dosage and temperature on the DPPH radical scavenging activity; (**B**). effects of enzyme dosage and pH on the DPPH radical scavenging activity; and (**C**). effects of temperature and pH on the DPPH clearance rate.

**Figure 3 molecules-28-06887-f003:**
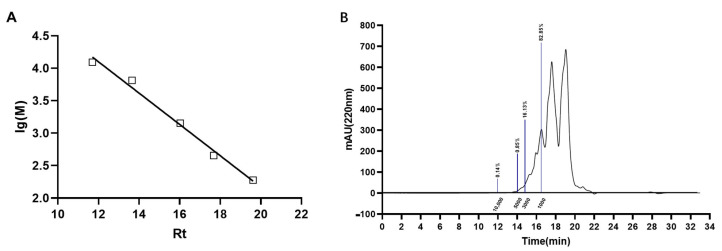
Relative molecular mass of enzymatic product. (**A**). Peptide relative molecular mass standard curve; Rt: retention time; lg(M): logarithm of MW. (**B**) Elution peak of enzymatic digestion product.

**Figure 4 molecules-28-06887-f004:**
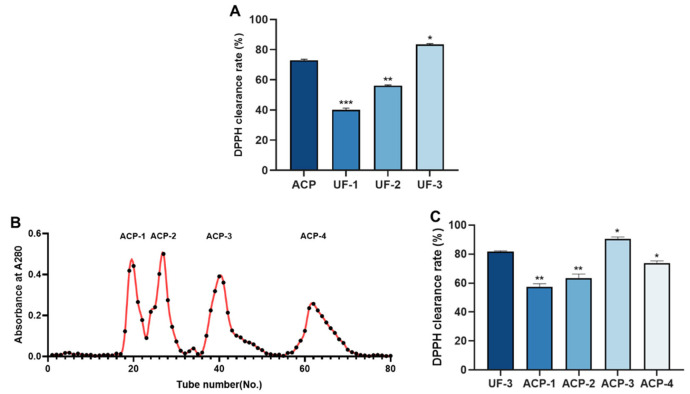
Isolation and purification of ACP. (**A**). DPPH clearance rate of peptides with four different molecular weights; (**B**). size exclusion chromatography of UF-3 fraction; (**C**). DPPH clearance rate of UF-3 and its fractions (ACP-1 to ACP-4). All results were triplicates of the mean ± SD. * *p* < 0.05, ** *p* < 0.01, *** *p* < 0.001.

**Figure 5 molecules-28-06887-f005:**
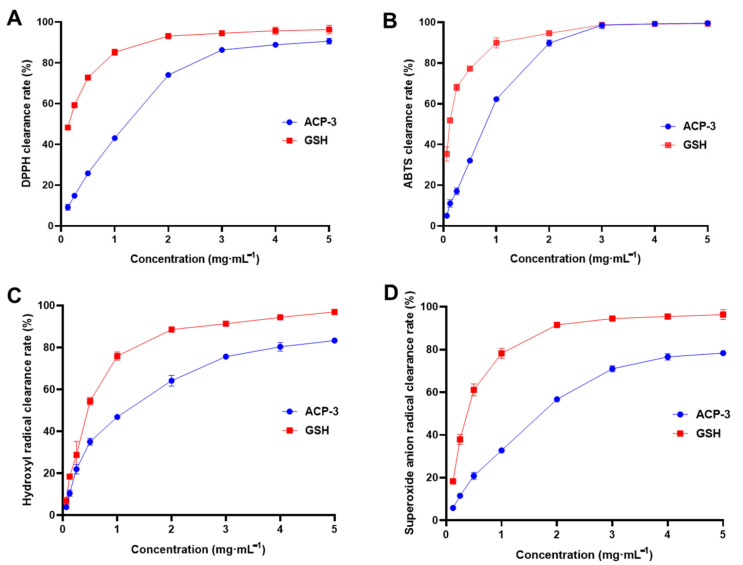
DPPH (**A**), ABTS (**B**), hydroxyl radical (**C**), and superoxide anion (**D**) clearance rate of ACP-3. GSH: glutathione. All results were triplicates of the mean ± SD.

**Figure 6 molecules-28-06887-f006:**
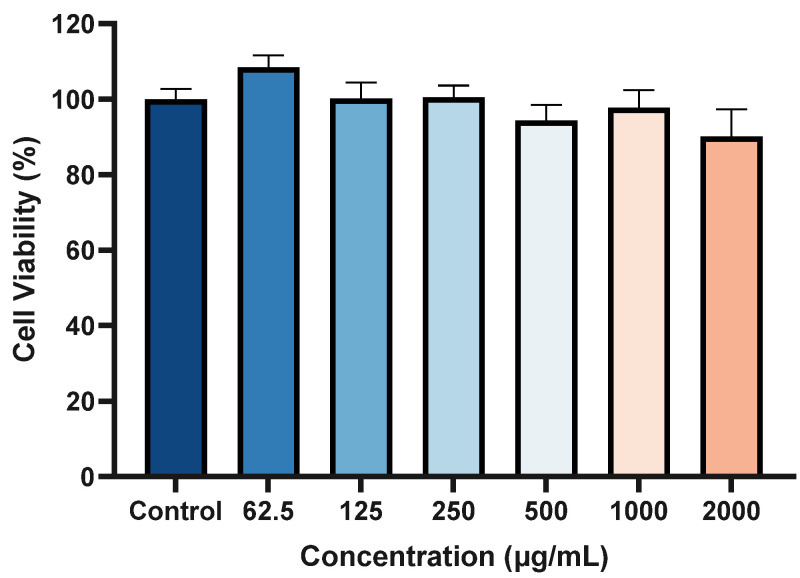
Cytotoxicity of ACP-3 at different concentrations on HaCaT cells. All results were triplicates of the mean ± SD.

**Figure 7 molecules-28-06887-f007:**
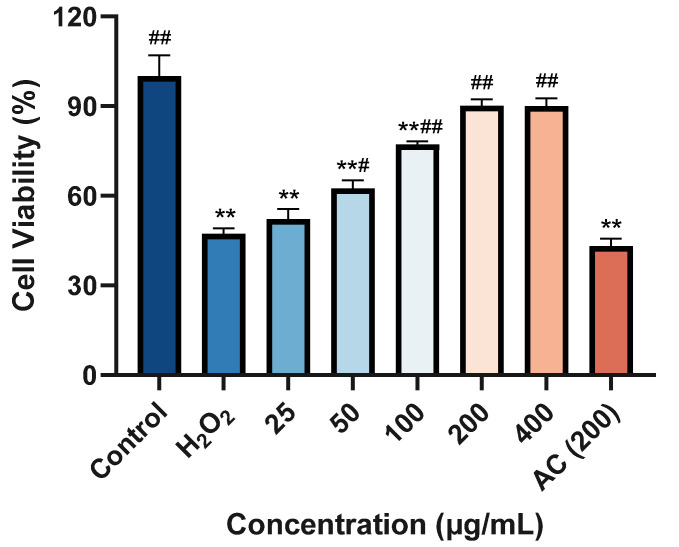
Effect of ACP-3 from antler ossified tissue on H_2_O_2_-induced damage HaCaT cells viability. All results were triplicates of the mean ± SD. ** *p* < 0.01 versus control; # *p* < 0.05, ## *p* < 0.01 versus H_2_O_2_ model group.

**Figure 8 molecules-28-06887-f008:**
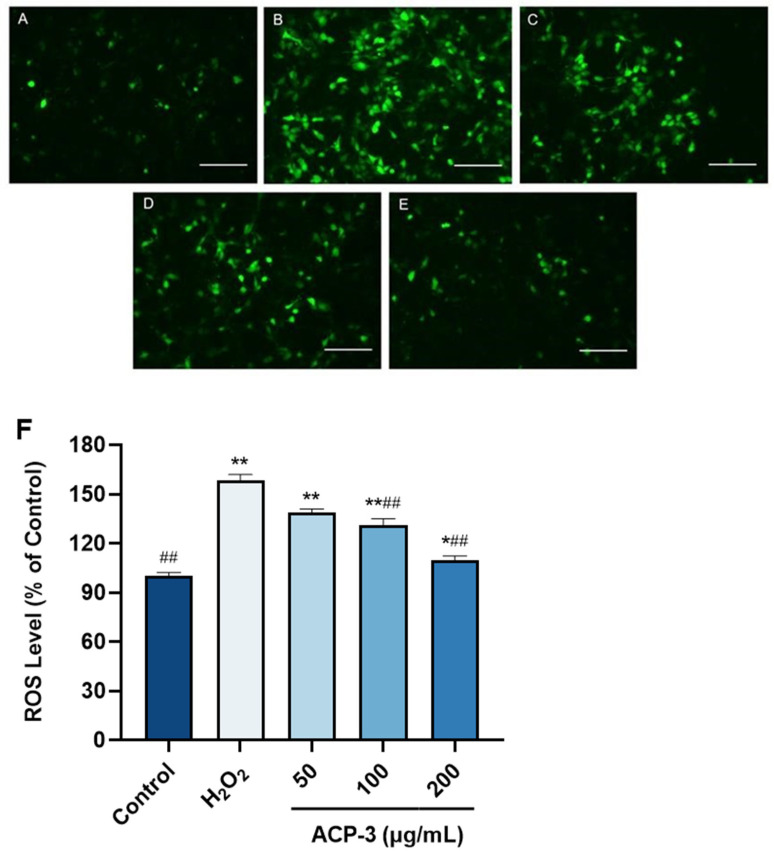
Fluorescence images of HaCaT cells treated with ACP-3 and H_2_O_2_. (**A**). Control; (**B**). 500 μM H_2_O_2_-injured; (**C**–**E**) H_2_O_2_-injured pretreated with 50, 100, 200 μg/mL ACP-3; (**F**) Effect of ACP-3 on the level of ROS in HaCaT cells with H_2_O_2_-induced damage. All results were triplicates of the mean ± SD. * *p* < 0.05, ** *p* < 0.01 versus control; ## *p* < 0.01 versus H_2_O_2_ damage group.

**Figure 9 molecules-28-06887-f009:**
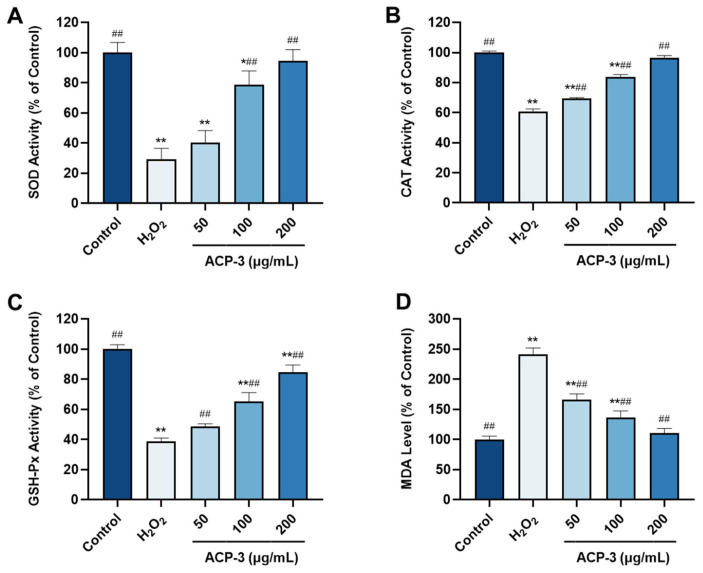
Effect of ACP-3 at different concentrations on levels of SOD (**A**), CAT (**B**), GSH-Px (**C**), and MDA (**D**) in H_2_O_2_-induced HaCaT cell damage model. All results were triplicates of the mean ± SD. * *p* < 0.05, ** *p* < 0.01 versus control; ## *p* < 0.01 versus H_2_O_2_ damage group.

**Table 1 molecules-28-06887-t001:** Response surface experimental design and results.

Run Numbers	A Enzyme Concentration (U/g)	B Temperature (°C)	C pH	R (DPPH Clearance Eate)
1	0 (3000)	−1 (45)	−1 (5.5)	63.84
2	−1 (2000)	1 (55)	0 (6.0)	70.33
3	0 (3000)	1 (55)	1 (6.5)	71.48
4	0 (3000)	−1 (45)	1 (6.5)	67.85
5	0 (3000)	0 (50)	0 (6.0)	73.21
6	0 (3000)	0 (50)	0 (6.0)	72.69
7	0 (3000)	0 (50)	0 (6.0)	73.55
8	0 (3000)	0 (50)	0 (6.0)	72.97
9	−1 (2000)	0 (50)	1 (6.5)	70.76
10	0 (3000)	0 (50)	0 (6.0)	73.38
11	0 (3000)	1 (55)	−1 (5.5)	69.43
12	−1 (2000)	−1 (45)	0 (6.0)	65.24
13	1 (4000)	1 (55)	0 (6.0)	72.33
14	1 (4000)	−1 (45)	0 (6.0)	67.03
15	1 (4000)	0 (50)	1 (6.5)	70.14
16	−1 (2000)	0 (50)	−1 (5.5)	65.92
17	1 (4000)	0 (50)	−1 (5.5)	69.52

**Table 2 molecules-28-06887-t002:** Analysis of variances for the developed regression equation.

Variables	Sum of Squares	*df*	Mean Square	F-Value	*p*-Value
Model	149.27	9	16.59	152.99	<0.0001
A (Enzyme concentration)	5.73	1	5.73	52.85	0.0002
B (Temperature)	48.02	1	48.02	442.96	<0.0001
C (pH)	16.62	1	16.62	153.29	<0.0001
AB	0.011	1	0.011	0.1	0.7591
AC	4.45	1	4.45	41.07	0.0004
BC	0.95	1	0.95	8.77	0.0211
A^2^	12.84	1	12.84	118.4	<0.0001
B^2^	30.32	1	30.32	279.69	<0.0001
C^2^	22.88	1	22.88	211.04	<0.0001
Residual	0.76	7	0.11		
Lack of Fit	0.3	3	0.1	0.88	0.5211
Pure Error	0.46	4	0.11		
Cor Total	150.03	16			
R^2^ = 0.9949			R_adj_^2^ = 0.9884		

**Table 3 molecules-28-06887-t003:** Amino acid composition of ACP-3.

Amino Acids	Proportion (%)
Aspartic acid (Asp) ^2^	6.609634551
Threonine (Thr) *	2.757641196
Serine (Ser)	3.718438538
Glutamic (Glu) ^2^	10.82292359
Glycine (Gly)	14.81146179
Alanine (Ala) ^1^	8.436710963
Cystine (Cys)	0.620431894
Valine (Val) *	2.993687708
Methionine (Met) *^, 1^	2.101328904
Isoleucine (Ile) *^, 1^	3.178571429
Leucine (Leu) *^, 1^	5.138870432
Tyrosine (Tyr)	1.243853821
Phenylalanine (Phe) *^,1^	2.56910299
Histidine (His) ^3^	1.174916944
Lysine (Lys) *^,3^	4.906146179
Arginine (Arg) ^3^	8.487541528
Proline (Pro) ^1^	20.48172757

*: essential amino acids; ^1^: hydrophobic amino acids; ^2^: acidic amino acids; ^3^: basic amino acids.

**Table 4 molecules-28-06887-t004:** Optimal reaction conditions of proteases.

Protease	Temperature (°C)	pH	Time (h)	Enzyme Concentration (U/g)	Concentration of Substrate (%)
Neutrase	50	6.5	4	2000	20
Alcalase	45	9.0	4	2000	20
Papain	55	6.0	4	2000	20
Trypsin	50	8.0	4	2000	20
Flavourzyme	50	7.5	4	2000	20

**Table 5 molecules-28-06887-t005:** Factors and levels in the response surface design.

Levels	Variables		
	A Enzyme Concentration (U/g)	B Temperature (°C)	C pH
−1	2000	45	5.5
0	3000	50	6
1	4000	55	6.5

## Data Availability

All relevant data about this research can be requested from the corresponding author.

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
