# Peer review of "Antioxidant Peptides from the Collagen of Antler Ossified Tissue and Their Protective Effects against H2O2-Induced Oxidative Damage toward HaCaT Cells"

_molecules, 2023, doi:10.3390/molecules28196887_

Round 1
Reviewer 1 Report
In this paper, the authors present a method for obtaining peptides with strong antioxidant activity. After isolation and purification, they describe, characterize, and test the technology in detail, providing clear and well-organized information. However, there are some comments that need to be addressed by the authors.
(1) The manuscript should be numbered correctly according to sections and sub-sections.
(2) The resolution of all figures should be improved.
(3) In Figure 1A, the wild-type AC protein should be tested for DPPH clearance as a control together with enzymatic hydrolysates. The concentration and positive control should also be clearly indicated.
(4) More details should be provided for statistical analysis, and significant differences between groups in all figures and legends should be identical.
(5) In the "2.4. Isolation and purification of ACP" section, each fraction yield should be discussed.
(6) What is GSH mean in Figure 5? The author should provide more information in the discussion and in the figure legend.
(7) A concentration of wild-type AC protein (such as 200 μg/ml) should be assayed in Figures 6-8 as a control to compare the differences between AC and ACP-3.
(8) More details about collagen extraction from antler from other researchers should be provided in the introduction and discussed.
Author Response
Dear Reviewer,
Thank you so much for your timely review for our manuscript. We appreciate you for the comments concerning our manuscript entitled “Antioxidant peptides from the collagen of antler ossified tissue and their protective effects against H2O2-Induced Oxidative Damage toward HaCaT cells” (molecules-2561194). The comments are all valuable and very helpful for revising and improving our paper, and give us the important guiding significance to our researches. We have studied the comments carefully and have made corrections which we hope to meet with approval. Revised portion are marked in red in the paper. Our responses are provided in the document "Reviewer 1". Please see the attachment.

Reviewer 2 Report
Manuscript ‘Antioxidant peptides from the collagen of antler ossified tissue and their protective effects against H2O2-Induced Oxidative Damage toward HaCaT cells’ by Chen et al., isolated a peptide from antler ossified tissue and analysed the antioxidant activity in Hacat cell line.
1. Since authors have mentioned about the possible future application in food processing and pharma, toxicity of the active antioxidant peptide needs to be evaluated.
2. Authors can discuss about the possible mechanism of action of ACP-3
Author Response
Dear Reviewer,
Thank you so much for your timely review for our manuscript. We appreciate you for the comments concerning our manuscript entitled “Antioxidant peptides from the collagen of antler ossified tissue and their protective effects against H2O2-Induced Oxidative Damage toward HaCaT cells” (molecules-2561194). The comments are all valuable and very helpful for revising and improving our paper, and give us the important guiding significance to our researches. We have studied the comments carefully and have made corrections which we hope to meet with approval. Revised portion are marked in red in the paper. Our responses are provided in the document "Reviewer 2". Please see the attachment.

Round 2
Reviewer 1 Report
This article has been revised as per my request, and I have no further comments.